# *Helicobacter pylori* eradication improves motor fluctuations in advanced Parkinson's disease patients: A prospective cohort study (HP-PD trial)

**Praween Lolekha**[1]*, **Thanakarn Sriphanom**[1¤], **Ratha-Korn Vilaichone**[2,3,4]

**1** Faculty of Medicine, Division of Neurology, Department of Internal Medicine, Thammasat University, Pathumthani, Thailand, **2** Faculty of Medicine, Department of Internal Medicine, Gastroenterology Unit, Thammasat University, Pathumthani, Thailand, **3** Department of Medicine, Chulabhorn International College of Medicine (CICM), Thammasat University, Pathumthani, Thailand, **4** Faculty of Medicine, Division of Gastroentero-Hepatology, Department of Internal Medicine, Universitas Airlangga, Surabaya, Indonesia

¤ Current address: Division of Neurology, Department of Internal Medicine, Lampang Provincial Hospital, Lampang, Thailand
* pwlolekha@gmail.com

**Data Availability Statement:** All relevant data are within the manuscript and its Supporting Information files.

## Abstract

### Background

*Helicobacter pylori* (HP) is a bacterium associated with many gastrointestinal (GI) diseases and has shown a high prevalence in Parkinson's disease (PD). As HP-associated GI dysfunction could affect L-dopa (levodopa) absorption, HP eradication might improve the clinical response and decrease motor fluctuations.

### Methods

A prospective cohort study was conducted on the clinical symptoms of PD patients with motor fluctuations. The $^{13}$C-urea breath test was used to diagnose a current HP infection. All patients with HP infection received a 2-week regimen of triple therapy. The changes in the Unified Parkinson's Disease Rating Scale (UPDRS) motor score, L-dopa onset time, wearing-off symptoms, mean daily on-off time, GI symptom scores, and quality of life score were measured at baseline and at a 6-week follow-up.

### Results

A total of 163 PD patients were assessed, of whom 40 were enrolled. Fifty-five percent of the enrolled patients (22/40) had a current HP infection, whereas HP eradication was identified in 17 of 22 (77.3%) patients who received eradication therapy. Patients with HP eradication showed a significant decrease in daily 'off' time (4.0 vs. 4.7 h, $p = 0.040$) and an increase in daily 'on' time (11.8 vs. 10.9 h, $p = 0.009$). Total wearing-off score (4.4 vs. 6.0, $p = 0.001$) and the GI symptom score (8.1 vs. 12.8, $p = 0.007$) were significantly improved. There was no significant improvement in L-dopa onset time, UPDRS motor score, or quality of life score.

**Funding:** This study was supported by a grant from Thammasat University Research Fund (TU Research Scholar, Contract No. 2/4/2562) and Thai Parkinson's Disease and Movement Disorder Society (Thai-PDMDS). The funders had no role in study design, data collection and analysis, decision to publish, or preparation of the manuscript.

**Competing interests:** The 13C-urea breath test and infrared spectrophotometer used in the study was provided by Thai Otsuka Pharmaceutical Co., Ltd. This does not alter our adherence to PLOS ONE policies on sharing data and materials.

## Conclusions

HP eradication leads to significant clinical improvement in the symptoms of PD. Eradication of HP not only increases the total daily 'on' time but also decreases wearing-off symptoms and improves GI symptoms.

## Introduction

Parkinson's disease (PD) is one of the most common neurodegenerative disorders that commonly affect elderly people. Despite enormous advanced knowledge of PD, L-dopa (levodopa) remains the most effective treatment for PD. Almost all PD patients require varying doses of L-dopa to manage their motor symptoms and maintain an acceptable quality of life. However, chronic L-dopa use in PD is associated with motor complications, including motor fluctuations and L-dopa-induced dyskinesia. After 5 years of L-dopa exposure, more than half of PD patients may experience some degree of these problems that significantly affects their functionality and quality of life [1]. Although PD mainly affects the motor system, it has been well established that PD is also associated with several non-motor symptoms (NMS). Gastrointestinal (GI) dysfunction is one of the most problematic NMS in PD and may precede the appearance of motor symptoms by several years. Recent evidence suggested an integral role of the microbiota-gut-brain axis for the early pathogenesis and progression of PD [2–5]. Also, several studies have shown an increased risk of PD in populations with chronic constipation and inflammatory bowel disease [6–8].

*Helicobacter pylori* (HP) is a Gram-negative bacterium that characteristically causes sustained inflammation of the gastric mucosa, which is associated with many GI diseases such as peptic ulcer, gastroduodenitis and fatal GI diseases especially gastric cancer. A recent population-based study has shown an increased risk of PD in chronic HP infection [9]. Also, meta-analysis has shown a significantly higher prevalence of HP infection in PD patients; furthermore, HP-infected PD patients demonstrated poorer control of motor symptoms than non-infected patients [10]. As GI dysfunction and L-dopa bioavailability play an important role in the pathophysiological changes underlying motor fluctuations, the eradication of HP might positively influence clinical outcome in the treatment of PD. This study aimed to evaluate the therapeutic effects of HP eradication on motor fluctuations and clinical symptoms in patients with advanced PD.

## Materials and methods

### Study design

A prospective cohort study was conducted between January and December 2019 at the Movement Disorders Clinic, Thammasat University Hospital, Thailand. The study was designed as a single-arm, open-label trial to evaluate the therapeutic effects of HP eradication in PD patients with motor fluctuations. The estimated sample size was determined according to the previous studies [11–14] with a level of significance of 5%, the statistical power of 80%, and the effect size of 0.8. The calculated number of HP infected patients needed in this study was 15 patients. Given the prevalence of HP infection in the general Thai population of 45% and the possibility of a 15% drop out rate, a total of 40 advanced PD patients were recruited. Consecutive PD patients were screened by a movement disorder specialist. Clinical assessment took place at baseline and 4 weeks after the 2-week HP eradication regimen period. Outcome

assessors were blinded to the HP results. The primary outcomes were measurable changes in daily on-off time, L-dopa onset and peak time that result from successful HP eradication. The secondary outcomes were changes in clinical symptoms and the quality of life scores. All participants were verbally explained regarding the study before signing the consent form. Written informed consent was obtained from all subjects prior to enrolment. The study protocol was approved by the local ethics committee of Thammasat University (MTU-EC-IM-6-179/61) and registered with the Thai Clinical Trials Registry (TCTR 20190222005).

## Participants

Inclusion criteria for the study population were age over 18 years, diagnosis of idiopathic PD according to the UK Parkinson's Disease Society Brain Bank Diagnostic Criteria (UKPDSBB), current presence of motor fluctuations and receiving treatment with L-dopa. Exclusion criteria were severe cognitive dysfunction or dementia, severe dysphagia, history of gastric lesions or prior gastric surgery, prior HP eradication therapy and allergic to amoxicillin, clarithromycin or omeprazole.

## Clinical assessment

Baseline characteristic data, taken from face-to-face interviews and medical records, were collected by a neurologist. Activities of daily living (ADL), NMS, GI symptoms, quality of life, wearing-off symptoms and daily motor fluctuations were assessed by the Schwab and England Activities of Daily Living Scale (SE-ADL), Non-Motor Symptom Questionnaire (NMSQ) application [15], seven-item GI complaint score (GI score), Parkinson's disease questionnaire-8 (PDQ-8), a Thai version of the 9-item Wearing-off Questionnaire (TWOQ-9) [16], and home PD diary, respectively. Disease duration and L-dopa equivalent dose (LED) were recorded. Motor impairment was examined by a neurologist and a movement disorder specialist during 'off' and 'on' periods with the Unified Parkinson's Disease Rating Scale (UPDRS), the Modified Hoehn and Yahr Scale (H&Y) and the Time Up and Go (TUG) test. Patients were asked to withhold all dopaminergic medications and fasting overnight ($\geq$ 8 h) for the morning 'off' assessment. Then the patients were given the regular morning dose of L-dopa. Both 'onset time' and 'peak time' of L-dopa effects were reported subjectively by the patients at the clinic. The 'on' motor assessments were done during the reported peak time or at 90 min after oral L-dopa intake, whichever comes earlier.

## Evaluation of a current HP infection

All patients underwent the $^{13}$C-urea breath test (Otsuka Pharmaceutical Co., Ltd, Tokyo, Japan) during their fasting, before taking L-dopa, and in the upright position. Patients must be free from antibiotics for at least 4 weeks and discontinue H2-receptor antagonist or proton pump inhibitors for at least 2 weeks before the HP assessment. The breath samples were collected using a special breath-collecting bag at baseline and 20 min after swallowing a 100-mg tablet of $^{13}$C-labelled urea. The breath samples were measured for isotope-labelled carbon dioxide ratio using infrared spectrophotometry. A cut-off delta value of $\geq$ 2.5% difference compared with pre- and post-dose breath samples is considered a positive result for current HP infection. The test has a reported sensitivity of 97.7% and a specificity of 98.4% [17].

## HP eradication therapy and outcome assessment

All patients continued with their usual dose of dopaminergic medication. The HP positive patients additionally received standard triple therapy with amoxicillin (1 mg twice daily), clarithromycin (500 mg twice daily) and omeprazole (40 mg twice daily) for 2 weeks.

Patients were not allowed to adjust or change their medication regimen. All patients received reminder phone calls for the follow-up date, 4 weeks after the 2-week regimen HP treatment. The Clinical Global Impression Scale (CGI) was assessed. HP eradication was defined as a negative $^{13}$C-urea breath test at least 4 weeks after treatment.

## Statistical analysis

Statistical analysis was performed using SPSS version 17.0 (SPSS Inc., Chicago, IL, USA). Descriptive statistical values were presented as mean ± standard deviation (SD) and percentage values. The independent *t*-test, Mann-Whitney test or chi-squared test was used to assess differences in various parameters between HP positive and HP negative groups. The dependent sample *t*-test or Wilcoxon matched-pairs signed rank test was used to assess the different outcomes between pre- and post-HP eradication in the HP positive group. Correlation analysis and multiple linear regression analysis were used to assess the associated factors of HP infection. All tests were two-sided and a probability value of $p < 0.05$ was considered to be statistically significant.

## Results

### Baseline characteristics and comparison

A total of 163 PD patients were consecutively assessed for eligibility. Of these, 123 patients were excluded, and 40 patients were enrolled in the study (Fig 1). The baseline characteristics

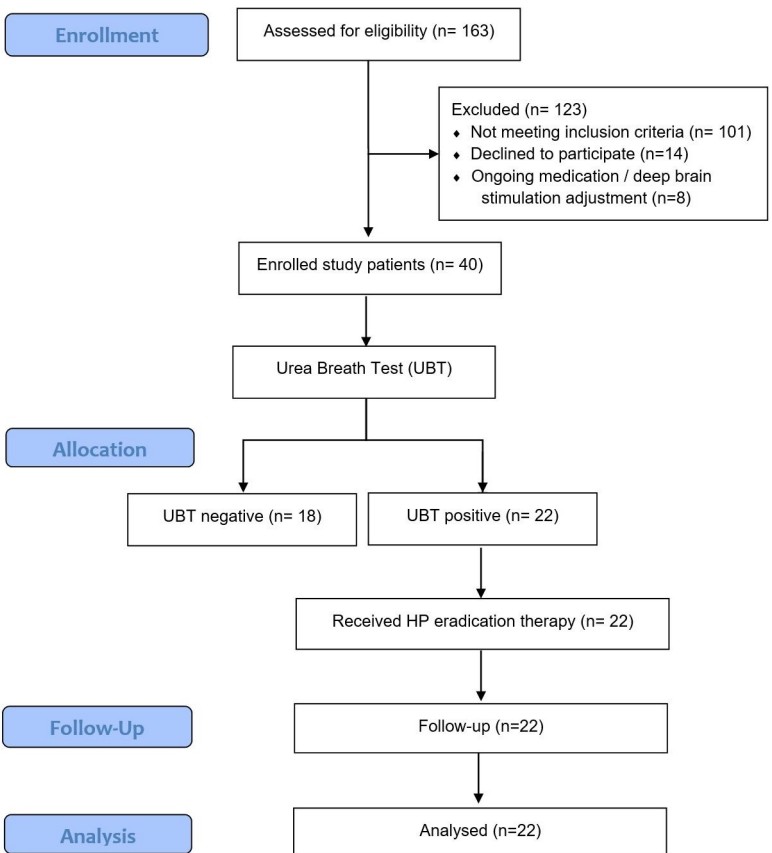

**Fig 1. Flow diagram of patient selection process.**

of the enrolled patients are shown in Table 1. All patients received L-dopa treatment, dopamine agonist (77.5%), catechol-O-methyltransferase (COMT) inhibitor (52.5%) and monoamine oxidase B (MAOB) inhibitor (10.0%). The mean LED was 740.70 ± 310.94 mg/day. Two patients were currently on deep brain stimulation (5.0%).

Twenty-two patients (55.0%) had a current HP infection. The parameters at baseline between HP positive and HP negative groups are shown in Table 1. There was no statistically

**Table 1. General demographic data of the study population.**

| | Total (*n* = 40) | HP positive (*n* = 22) | HP negative (*n* = 18) | *p* |
|---|---|---|---|---|
| Male gender, *n* (%) | 20 (50.0) | 12 (60.0) | 8 (40.0) | 0.525 |
| Age (years) | 63.6 ± 9.6 | 60.2 ± 10.3 | 67.8 ± 6.8 | **0.010** |
| Age at onset (years) | 55.9 ± 9.6 | 52.8 ± 10.0 | 59.7 ± 8.0 | **0.023** |
| Smoking, *n* (%) | 2 (5.0) | 2 (9.1) | 0 (0.0) | 0.176 |
| Vegetable/fruit (servings/day) | 2.3 ± 0.7 | 2.2 ± 0.8 | 2.5 ± 0.6 | 0.173 |
| Disease duration (years) | 7.9 ± 5.0 | 7.5 ± 4.4 | 8.4 ± 5.8 | 0.542 |
| SE-ADL | 80.0 ± 6.4 | 77.7 ± 4.3 | 82.8 ± 7.5 | **0.018** |
| Hoehn & Yahr stage | 2.2 ± 0.7 | 2.2 ± 0.6 | 2.2 ± 0.9 | 0.966 |
| Total LED (mg/day) | 740.7 ± 310.9 | 738.8 ± 289.3 | 743.1 ± 344.1 | 0.966 |
| • Dopamine agonist, *n* (%) | 31 (77.5) | 19 (86.4) | 12 (66.7) | 0.144 |
| • COMT inhibitor, *n* (%) | 21 (52.5) | 10 (45.5) | 11 (61.1) | 0.336 |
| • MAO-B inhibitor, *n* (%) | 4 (10.0) | 2 (9.1) | 2 (11.1) | 0.839 |
| • Anticholinergic use, *n* (%) | 8 (20.0) | 7 (31.8) | 1 (5.6) | **0.039** |
| Total TWOQ-9 score | 5.3 ± 1.8 | 5.7 ± 1.5 | 4.7 ± 2.0 | 0.092 |
| • Tremor, *n* (%) | 26 (65.0) | 18 (81.8) | 8 (44.0) | **0.014** |
| • Anxiety, *n* (%) | 10 (25.0) | 7 (31.9) | 3 (16.7) | 0.271 |
| • Mood change, *n* (%) | 11 (27.5) | 8 (36.4) | 3 (16.7) | 0.165 |
| • Slowness, *n* (%) | 38 (95.0) | 21 (95.5) | 17 (94.4) | 0.884 |
| • Reduced dexterity, *n* (%) | 33 (82.5) | 20 (90.9) | 13 (72.2) | 0.122 |
| • Stiffness, *n* (%) | 26 (65.0) | 15 (68.2) | 11 (61.1) | 0.641 |
| • Slowness of thinking, *n* (%) | 16 (40.0) | 8 (36.4) | 8 (44.4) | 0.604 |
| • Muscle cramping, *n* (%) | 25 (62.5) | 13 (59.1) | 12 (66.7) | 0.622 |
| • Pain/aching, *n* (%) | 25 (62.5) | 16 (72.8) | 9 (50.0) | 0.140 |
| Total GI symptom score | 13.7 ± 7.4 | 12.4 ± 6.7 | 15.4 ± 8.1 | 0.205 |
| Total NMSQ score | 35.4 ± 17.1 | 33.8 ± 13.5 | 37.9 ± 20.9 | 0.425 |
| PDQ-8 score | 9.9 ± 5.8 | 10.4 ± 4.9 | 9.2 ± 6.8 | 0.526 |
| 'Off' UPDRS motor score | 30.9 ± 12.2 | 31.7 ± 9.9 | 29.8 ± 14.8 | 0.653 |
| 'On' UPDRS motor score | 14.7 ± 9.3 | 14.3 ± 7.5 | 15.1 ± 11.3 | 0.800 |
| Time up and go (sec) | 12.0 ± 7.4 | 11.9 ± 8.3 | 12.1 ± 6.3 | 0.942 |
| L-Dopa onset time (min) | 19.2 ± 9.5 | 22.1 ± 8.1 | 15.4 ± 10.0 | **0.025** |
| L-Dopa peak time (min) | 35.0 ± 9.5 | 35.4 ± 8.8 | 34.5 ± 10.5 | 0.785 |
| Daily 'on' time (h) | 11.4 ± 2.2 | 10.9 ± 1.9 | 12.0 ± 2.4 | 0.117 |
| Daily 'off' time (h) | 4.3 ± 1.9 | 4.6 ± 1.6 | 3.9 ± 2.2 | 0.211 |
| Dyskinesia (h) | 1.4 ± 2.1 | 1.3 ± 1.8 | 1.5 ± 2.3 | 0.738 |

HP: *Helicobacter pylori*; SE-ADL: Schwab and England Activities of Daily Living Scale; LED: Levodopa equivalent dose; COMT: Catechol-O-methyltransferase; MAO, monoamine oxidase; TWOQ-9: Thai version of the 9-item Wearing-off Questionnaire; GI: Gastrointestinal; NMSQ: Non-Motor Symptom Questionnaire; PDQ: Parkinson's Disease Questionnaire; UPDRS: Unified Parkinson's Disease Rating Scale; L-Dopa: Levodopa.

Statistically significant *p*-values are in bold.

significant difference in the mean daily LED between groups. Patients with HP infection had statistically significantly younger age (60.2 vs. 67.8 years, *p* = 0.010) and age of onset (52.8 vs. 59.7 years, *p* = 0.023), poorer SE-ADL (77.7 vs. 82.8, *p* = 0.018), a higher rate of anticholinergic use (31.8 vs. 5.6%, *p* = 0.039), a higher rate of wearing-off tremor (81.8 vs. 44%, *p* = 0.014) and delayed L-dopa onset time (22.1 vs. 15.4 min, *p* = 0.025) than the non-HP-infected patients. Multiple linear regression analysis showed significant association between levodopa onset time (*p* = 0.014), SE-ADL scale (*p* = 0.038), and HP infection. The common GI symptoms in both groups were constipation (87.5%), bloating (60.0%) and abdominal pain (37.5%). There was no statistically significant difference in disease duration, dietary history, UPDRS motor scores, H&Y stage, TUG test, total TWOQ-9 scores, daily 'on' and 'off' times, PDQ-8 score, total NMSQ score or total GI symptom scores between groups.

## Analysis of HP eradication effects

HP eradication was identified in 17 patients (77.3%). Patients with confirmed HP eradication showed statistically significantly decreased daily 'off' time (4.0 vs. 4.7 h, *p* = 0.040) and increased total daily 'on' time (11.8 vs. 10.9 h, *p* = 0.009) by approximately 54 min per day from baseline. The success of HP eradication therapy significantly reduced daily 'off' time and increased 'on' time by 0.7 ± 1.3 and 0.9 ± 1.2 h, respectively, compared with a 0.6 ± 0.4 h increase in 'off' time and a 1.6 ± 2.3 h decrease in 'on' time with eradication failure (HP eradication effect: −1.3 off-hours, *p* = 0.041, and +2.5 on-hours, *p* = 0.004) (Fig 2). In addition, there were significant improvements in total TWOQ-9 score (4.4 vs. 6.0, *p* = 0.001), TWOQ-9 tremor subscore (0.5 vs. 0.8, *p* = 0.009), TWOQ-9 mood change subscore (0.1 vs. 0.4, *p* = 0.029), total GI symptom score (8.1 vs. 12.8, *p* = 0.007), bloating subscore (1.1 vs. 2.9, *p* = 0.007) and dysphagia subscore (0.3 vs. 0.7, *p* = 0.029) following successful eradication (Table 2). Multiple testing correction by Bonferroni adjustment showed a statistically significant improvement only in total TWOQ-9 score (corrected *p* = 0.029) (S1 Table). Moreover, there were trends toward improvement but not statistically significant in L-dopa onset time, UPDRS motor score, total NMSQ score, and PDQ-8 score. By contrast, patients who failed

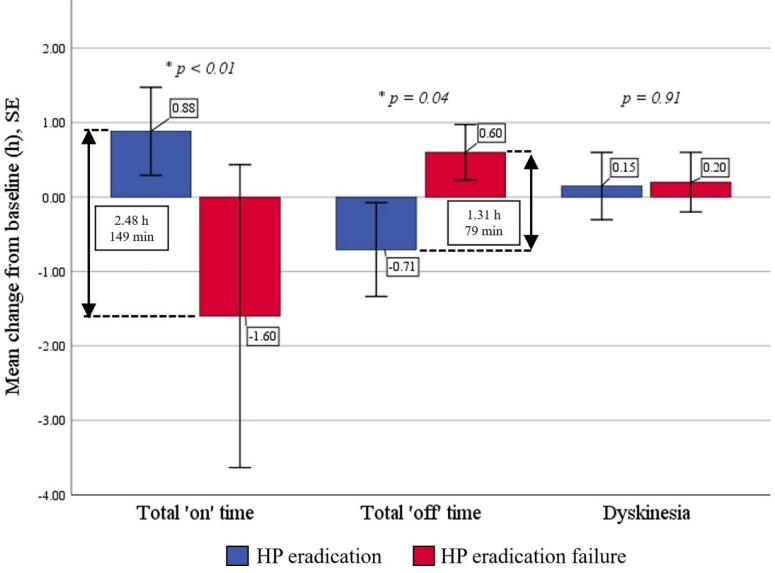

**Fig 2. Mean change from baseline for motor fluctuations between successful and failed *Helicobacter pylori* (HP) eradication groups.**

**Table 2. Clinical symptoms before and after successful *Helicobacter pylori* (HP) eradication (*n* = 17).**

| | Before HP eradication | After HP eradication | *p* |
|---|---|---|---|
| Total TWOQ-9 score | 6.0 ± 1.3 | 4.4 ± 1.7 | **0.001** |
| • Tremor | 0.8 ± 0.4 | 0.5 ± 0.5 | **0.009** |
| • Anxiety | 0.4 ± 0.5 | 0.1 ± 0.3 | 0.096 |
| • Mood changes | 0.4 ± 0.5 | 0.1 ± 0.2 | **0.029** |
| • Slowness | 0.9 ± 0.2 | 0.9 ± 0.2 | 1.000 |
| • Reduced dexterity | 0.9 ± 0.3 | 0.8 ± 0.4 | 0.163 |
| • Stiffness | 0.8 ± 0.4 | 0.6 ± 0.5 | 0.188 |
| • Slowness of thinking | 0.7 ± 0.5 | 0.5 ± 0.5 | 0.431 |
| • Muscle cramping | 0.8 ± 0.4 | 0.7 ± 0.5 | 0.431 |
| • Pain/aching | 0.8 ±0.4 | 0.7 ± 0.5 | 0.431 |
| Total GI symptom score | 12.8 ± 7.3 | 8.1 ± 4.5 | **0.007** |
| • Heartburn | 1.2 ± 2.8 | 0.6 ± 1.4 | 0.443 |
| • Bloating | 2.9 ± 3.2 | 1.1 ± 1.6 | **0.007** |
| • Nausea | 0.2 ± 0.5 | 0.2 ± 0.5 | 1.000 |
| • Vomiting | 0.1 ± 0.5 | 0.1 ± 0.5 | 1.000 |
| • Abdominal pain | 1.8 ± 2.6 | 0.9 ± 2.2 | 0.112 |
| • Diarrhea | 0.4 ± 1.1 | 0.0 ± 0.0 | 0.130 |
| • Constipation | 6.2 ± 3.5 | 5.2 ± 2.8 | 0.146 |
| Total NMSQ score | 33.4 ± 13.3 | 29.3 ± 14.1 | 0.258 |
| • Dysphagia | 0.7 ± 1.0 | 0.3 ± 0.6 | **0.029** |
| PDQ-8 score | 10.9 ± 5.3 | 8.6 ± 5.2 | 0.080 |
| 'Off' UPDRS motor score | 29.9 ± 9.6 | 26.0 ± 10.0 | 0.104 |
| 'On' UPDRS motor score | 12.7 ± 7.0 | 10.6 ± 5.0 | 0.185 |
| Time up and go (sec) | 11.8 ± 8.5 | 9.8 ±2.8 | 0.242 |
| L-Dopa onset time (min) | 21.2 ± 6.9 | 19.5 ± 9.6 | 0.468 |
| L-Dopa peak time (min) | 35.7 ± 7.9 | 34.9 ± 13.8 | 0.782 |
| Daily 'on' time (h) | 10.9 ± 1.9 | 11.8 ± 2.0 | **0.009** |
| Daily 'off' time (h) | 4.7 ± 1.5 | 4.0 ± 2.1 | **0.040** |
| Dyskinesia (h) | 1.1 ± 1.6 | 1.3 ± 1.8 | 0.524 |

TWOQ-9: Thai version of the 9-item Wearing-off Questionnaire; GI: Gastrointestinal; NMSQ: Non-Motor Symptoms Questionnaire; PDQ: Parkinson's Disease Questionnaire; UPDRS: Unified Parkinson's Disease Rating Scale; L-Dopa: Levodopa.

Statistically significant *p*-values are in bold.

eradication therapy did not demonstrate any benefit after treatment but showed significantly increased daily 'off' time (4.9 vs. 4.3 h, *p* = 0.033) (Table 3).

Compared with baseline status in HP negative group, the success of HP eradication therapy demonstrated a significant lower total GI symptom score (15.4 vs. 8.1, *p* = 0.003), heartburn subscore (2.4 vs. 0.6, *p* = 0.024), bloating subscore (3.3 vs. 1.1, *p* = 0.007), and dysphagia subscore (1.3 vs. 0.3, *p* = 0.008). There was no significant difference in motor-related scores, L-dopa onset time, and quality of life between groups (Table 4).

In terms of side-effects related to HP eradication therapy, only one patient in the treatment failure group reported significant side effects (stomach pain, nausea and vomiting) to the treatment. Patients who failed eradication therapy were all male and had a significantly lower intake of vegetables and fruit (1.4 vs. 2.4 servings/day, *p* < 0.01) as well as a higher rate of smoking (40.0 vs. 0.0%, *p* < 0.01) compared with the HP eradication group. There was no statistically significant difference in age, age at onset, disease duration, SE-ADL scale, H&Y stage,

**Table 3. Clinical symptoms before and after receiving the eradication therapy in *Helicobacter pylori* (HP) eradication failure group (*n* = 5).**

| | Before HP eradication | After HP eradication | *p* |
|---|---|---|---|
| Total TWOQ-9 score | 4.8 ± 1.6 | 4.2 ± 2.2 | 0.529 |
| • Tremor | 0.8 ± 0.4 | 0.4 ± 0.5 | 0.178 |
| • Mood changes | 0.2 ± 0.4 | 0.4 ± 0.5 | 0.374 |
| Total GI symptom score | 10.8 ± 4.2 | 11.4 ± 1.3 | 0.763 |
| • Bloating | 1.8 ± 2.2 | 2.6 ± 2.1 | 0.338 |
| Total NMSQ score | 30.0 ± 14.9 | 25.2 ± 16.0 | 0.549 |
| PDQ-8 score | 8.6 ± 3.1 | 9.4 ± 6.4 | 0.759 |
| 'Off' UPDRS motor score | 33.0 ± 6.2 | 30.4 ± 8.2 | 0.610 |
| 'On' UPDRS motor score | 17.2 ± 7.6 | 12.0 ± 5.0 | 0.127 |
| Time up and go (sec) | 9.1 ± 2.6 | 8.3 ± 1.0 | 0.478 |
| L-Dopa onset time (min) | 23.7 ± 12.4 | 22.0 ± 6.7 | 0.645 |
| L-Dopa peak time (min) | 33.6 ± 12.8 | 28.9 ± 7.3 | 0.248 |
| Daily 'on' time (h) | 10.9 ± 2.3 | 9.3 ± 2.3 | 0.191 |
| Daily 'off' time (h) | 4.3 ± 2.3 | 4.9 ± 2.3 | **0.033** |
| Dyskinesia (h) | 1.6 ± 2.8 | 1.8 ± 3.3 | 0.374 |

TWOQ-9: Thai version of the 9-item Wearing-off Questionnaire; GI: Gastrointestinal; NMSQ: Non-Motor Symptoms Questionnaire; PDQ: Parkinson's Disease Questionnaire; UPDRS: Unified Parkinson's Disease Rating Scale; L-Dopa: Levodopa.

Statistically significant *p*-values are in bold.

UPDRS motor score, LED and antiparkinsonian medications between groups (Table 5). Patients with successful HP eradication reported significant clinical benefit on the CGI scale: 60% compared with 20% in the failure group.

## Discussion

The reported prevalence of HP infection in PD patients varied between 32% and 70% in different studies, depended on age group, gender, geographical origin, socioeconomic status of the studied population and methods for diagnosis [18]. This study found HP infection in 55% of Thai patients with advanced PD, which is slightly higher than finding prevalence rates of 40–50% among the Thai general population [12]. However, the prevalence in our study is much higher in the subgroup of patients aged 40–60 years than the age-matched general Thai population (78.6% vs. 48.1%) [19]. In contrast to a study by Tan et al. [20], our study demonstrated a statistically significantly younger age and age of onset of the HP positive group than the HP negative group. This could be due to the difference in study population, with our study mainly focusing on patients with advanced PD experiencing motor fluctuations. Also, young PD patients in our study had a significantly higher rate of anticholinergic use, which might be associated with a reduction in volume and acidity of gastric secretion as well as GI motility, resulting in an alteration of the normal flora that could contribute to bacterial colonization, overgrowth, and HP infection.

In terms of GI symptoms, our study did not demonstrate any difference between HP positive and HP negative groups. However, HP eradication significantly alleviated the symptoms of bloating and dysphagia in HP-infected PD patients. This suggested that GI symptoms in PD are not solely the effect of HP infection, but the HP infection could aggravate and induce chronic inflammation in the gastroduodenal mucosa, causing several GI symptoms. Dysphagia typically emerges in PD patients with advanced motor symptoms, and often related to gastro-esophageal reflux disease. Eradication of HP in our study did not significantly improve motor

**Table 4. Clinical symptoms between patients with *Helicobacter pylori* (HP) negative and successful HP eradication groups.**

| | HP negative (*n* = 18) | HP positive at baseline (*n* = 17) | | | |
| --- | --- | --- | --- | --- | --- |
| | | Pre-HP eradication | $p^a$ | Post-HP eradication | $p^a$ |
| Total TWOQ-9 score | 4.7 ± 2.0 | 6.0 ± 1.3 | **0.028** | 4.4 ± 1.7 | 0.625 |
| Total GI symptom score | 15.4 ± 8.1 | 12.8 ± 7.3 | 0.335 | 8.1 ± 4.5 | **0.003** |
| • Heartburn | 2.4 ± 2.8 | 1.2 ± 2.8 | 0.211 | 0.6 ± 1.4 | **0.024** |
| • Bloating | 3.3 ± 2.7 | 2.9 ± 3.2 | 0.740 | 1.1 ± 1.6 | **0.007** |
| • Nausea | 0.6 ± 1.5 | 0.2 ± 0.5 | 0.267 | 0.2 ± 0.5 | 0.267 |
| • Vomiting | 0.1 ± 0.5 | 0.1 ± 0.5 | 0.968 | 0.1 ± 0.5 | 0.968 |
| • Abdominal pain | 1.6 ± 2.5 | 1.8 ± 2.6 | 0.760 | 0.9 ± 2.2 | 0.404 |
| • Diarrhea | 0.5 ± 1.2 | 0.4 ± 1.1 | 0.819 | 0.0 ± 0.0 | 0.096 |
| • Constipation | 6.9 ± 3.5 | 6.2 ± 3.5 | 0.516 | 5.2 ± 2.8 | 0.115 |
| Total NMSQ score | 37.9 ± 20.9 | 33.4 ± 13.3 | 0.544 | 29.3 ± 14.1 | 0.173 |
| • Dysphagia | 1.3 ± 1.3 | 0.7 ± 1.0 | 0.124 | 0.3 ± 0.6 | **0.008** |
| PDQ-8 score | 9.2 ± 6.8 | 10.9 ± 5.3 | 0.409 | 8.6 ± 5.2 | 0.759 |
| 'Off' UPDRS motor score | 29.8 ± 14.8 | 29.9 ± 9.6 | 0.740 | 26.0 ± 10.0 | 0.388 |
| 'On' UPDRS motor score | 15.1 ± 11.3 | 12.7 ± 7.0 | 0.618 | 10.6 ± 5.0 | 0.146 |
| Time up and go (sec) | 12.1 ± 6.3 | 11.8 ± 8.5 | 0.817 | 9.8 ± 2.8 | 0.289 |
| L-Dopa onset time (min) | 15.4 ± 10.0 | 21.2 ± 6.9 | **0.041** | 19.5 ± 9.6 | 0.229 |
| L-Dopa peak time (min) | 34.5 ± 10.5 | 35.7 ± 7.9 | 0.669 | 34.9 ± 13.8 | 0.979 |
| Daily 'on' time (h) | 12.0 ± 2.4 | 10.9 ± 1.9 | 0.140 | 11.8 ± 2.0 | 0.781 |
| Daily 'off' time (h) | 3.9 ± 2.2 | 4.7 ± 1.5 | 0.181 | 4.0 ± 2.1 | 0.819 |
| Dyskinesia (h) | 1.5 ± 2.3 | 1.1 ± 1.6 | 0.630 | 1.3 ± 1.8 | 0.802 |

[a] Compared to HP negative group.

TWOQ-9: Thai version of the 9-item Wearing-off Questionnaire; GI: Gastrointestinal; NMSQ: Non-Motor Symptoms Questionnaire; PDQ: Parkinson's Disease Questionnaire; UPDRS: Unified Parkinson's Disease Rating Scale; L-Dopa: Levodopa.

Statistically significant *p*-values are in bold.

scores or reflux symptoms. However, HP positive patients demonstrated significant improvement of dysphagia subscore after eradication therapy. Also, the post-eradication total GI symptoms, bloating, heartburn, and dysphagia scores were significantly less than baseline scores of HP negative patients. This might suggest additional effects of eradication therapy, especially from a gastric acid-lowering agent on gastroesophageal reflux disease that might contribute to dysphagia in PD patients. It should be noted that patients with severe dysphagia were excluded from the study. The significant improvement of dysphagia score in our study was minimal and might not significantly be observed in clinical practice. A multidisciplinary approach and identification of modifiable factors and causes are keys to the success of GI management. Although 10–14 days of triple therapy is recommended as a first-line HP eradication method in Thailand, treatment failure might be observed in approximately 20% [21]. In our study, the HP eradication rate was 77.3%, which is slightly lower than that of previous reports in Thailand [21,22]. This might reflect the increasing rate of antimicrobial resistance of the microorganism, therefore the preferred eradication regimen should vary by region depending on the known or anticipated pattern of antimicrobial resistance [21]. Interestingly, patients who failed the eradication therapy had a significantly lower intake of fruit and vegetables per day and a higher rate of smoking. This might support the interaction between environmental components and the outcome of HP infection therapy.

Regarding motor symptoms, patients with HP infection had longer mean L-dopa onset time and a higher frequency of TWOQ-9 tremor compared to those without infection, which

**Table 5. Demographic and clinical baseline status between *Helicobacter pylori* (HP) positive patients who succeeded and failed the eradication therapy.**

| | Successful eradication (*n* = 17) | Failure eradication (*n* = 5) | *P* |
|---|---|---|---|
| Male gender, *n* (%) | 7 (41.2) | 5 (100.0) | **0.020** |
| Age (years) | 61.1 ± 10.6 | 57.4 ± 9.8 | 0.495 |
| Age at onset (years) | 53.3 ± 10.3 | 50.4 ± 9.5 | 0.544 |
| Smoking, n (%) | 0 (0.0) | 2 (40.0) | **0.006** |
| Vegetable/fruit (servings/day) | 2.4 ± 0.6 | 1.4 ± 0.9 | **0.009** |
| Disease duration (years) | 7.6 ± 4.2 | 7.0 ± 5.5 | 0.834 |
| SE-ADL | 77.1 ± 4.7 | 80 ± 0.0 | 0.184 |
| Hoehn & Yahr stage | 2.2 ± 0.6 | 2.2 ± 0.3 | 0.976 |
| 'Off' UPDRS motor score | 31.3 ± 10.9 | 33.0 ± 6.2 | 0.664 |
| 'On' UPDRS motor score | 13.5 ± 7.5 | 17.2 ± 7.6 | 0.370 |
| Total LED (mg/day) | 774.3 ± 312.7 | 617.9 ± 156.9 | 0.152 |
| • Dopamine agonist, n (%) | 14 (82.4) | 5 (100.0) | 0.312 |
| • COMT inhibitor, n (%) | 9 (52.9) | 1 (20.0) | 0.193 |
| • MAO-B inhibitor, n (%) | 1 (5.9) | 1 (20.0) | 0.334 |
| • Anticholinergic use, n (%) | 6 (35.3) | 1 (20.0) | 0.519 |

SE-ADL: Schwab and England Activities of Daily Living Scale; UPDRS: Unified Parkinson's Disease Rating Scale; LED: Levodopa equivalent dose; COMT: catechol-O-methyltransferase; MAO: Monoamine oxidase.

Statistically significant *p*-values are in bold.

could suggest an association between HP infection and L-dopa bioavailability. Although the L-dopa onset time was not changed after HP eradication, patients with successful HP eradication demonstrated a reduction in total TWOQ-9 score and daily 'off' time, as well as an increase in total daily 'on' time which consistent with the previous findings [11–14], but contrasts with a post hoc analysis of a recent randomized controlled trial study using a motor fluctuation score detected by wearable sensor [23]. Discrepancies with this report might be explained by the difference of the study population which mainly focus on stable PD without motor fluctuations. There were only 7 PD patients with motor fluctuations in the treatment group evaluated in this post hoc analysis [23]. In our study, HP eradication improved 'on' time by approximately 54 min per day which similar to a finding in the previous study of 56 min per day at 6 weeks [14], and comparable to adjunct therapy with 1 mg rasagiline in PD patients with mild motor fluctuations [24]. The prolonged "on" duration without changing in L-dopa onset and peak times in our study were consistent with the previous pharmacokinetic study that demonstrated a significantly higher of the area under plasma L-dopa concentration-time curve without a significant changing in time to maximum concentration after the HP eradication [11]. Therefore, the definite process by which HP may affect L-dopa absorption is still unknown. It was hypothesized that gastroduodenitis induced by HP infection could delay the gastric emptying time, causing delayed L-dopa delivery into the duodenum and impaired L-dopa active transport at the site of L-dopa absorption [11,14,25]. Moreover, HP might indirectly interrupt L-dopa absorption by producing urease, an enzyme that hydrolyses urea to ammonia and carbon dioxide, thus raising the gastric pH and possibly affecting the pH-sensitive L-dopa solubility [26]. Furthermore, HP might play a part in the biosynthesis of neurotoxins that directly affect dopaminergic neurons [27]. Eradication of HP should normalize the gastric pH and ameliorate L-dopa absorption, resulting in improvement of the clinical response [11,14]. Although our study did not demonstrate a significant benefit of HP eradication on L-dopa onset time, there were trends toward the improvement of L-dopa onset time as suggested in previous studies [12–14]. Similar to a recent randomized-control trial study [23], our study did not

demonstrate a significant benefit of HP eradication on UPDRS motor score, gait speed, and NMSQ score. These findings suggested that HP eradication should not be used as a symptomatic adjunct therapy in stable PD but might be considered as a treatment for motor fluctuations.

Strengths of this study were the study design that focuses on PD patients with motor fluctuations and the therapeutic effects of HP eradication on various GI symptoms. The outcome assessors were blinded to the patients' HP status both at baseline and after the eradication treatment. Also, the participants in our study had a high adherence with prescribed treatment. There were no dropouts during the 6-week follow-up.

We acknowledge that our present study has some limitations. First, our study was based on one center and the number of participants was limited. Even though the study population was adequate to estimate the effect of HP eradication on motor fluctuation, it might be too small to identify the definite prevalence and clinical symptoms of HP infection in PD patients. Also, there were multiple comparisons in the statistical analysis that might cause statistical significance occurred by chance. Second, our study was an open-label, single-arm, 6-week clinical trial. Thus, the benefit of HP eradication might contribute to the placebo effect, however this was not demonstrated in the treatment failure group. A multicenter, prospective, randomized controlled study with a larger sample size and long-term follow-up should be considered in the future. Third, some data such as L-dopa onset and peak and daily 'on' and 'off' times were based on subjective evaluation. An objective motor evaluation with home-based wearable measurements should be considered. Lastly, the benefits seen in our short-term study may not translate into improvements in the longer term. Also, variations in lifestyle factors such as dietary habits, the compositional changes of gut microbiota, and other associated factors might affect the disease progression or interfere with L-dopa absorption.

## Conclusions

Successful HP eradication therapy leads to significant clinical improvements in the symptoms of PD patients with motor fluctuations. Eradication of HP not only increases the total daily 'on' time but also decreases wearing-off symptoms and improves GI symptoms. HP eradication was well tolerated and should be considered as an additional treatment for motor fluctuations in PD.

## Supporting information

**S1 Checklist. TREND statement checklist.**
(PDF)

**S1 Table. Clinical symptoms before and after successful *Helicobacter pylori* (HP) eradication with corrected *p*-values.**
(DOCX)

**S1 Protocol. Study protocol (Thai).**
(PDF)

**S2 Protocol. Study protocol (English).**
(PDF)

## Acknowledgments

We thank the Digestive Diseases Research Centre (DRC) at Thammasat University Hospital for their assistance with the UBT tests. We would also like to show our gratitude to the patients and their families who took part and participated in the study.

## Author Contributions

**Conceptualization:** Praween Lolekha, Ratha-Korn Vilaichone.

**Data curation:** Thanakarn Sriphanom.

**Formal analysis:** Praween Lolekha, Thanakarn Sriphanom.

**Funding acquisition:** Praween Lolekha, Ratha-Korn Vilaichone.

**Investigation:** Praween Lolekha, Thanakarn Sriphanom.

**Methodology:** Praween Lolekha.

**Project administration:** Praween Lolekha.

**Supervision:** Praween Lolekha, Ratha-Korn Vilaichone.

**Validation:** Praween Lolekha.

**Writing – original draft:** Praween Lolekha, Thanakarn Sriphanom.

**Writing – review & editing:** Praween Lolekha, Ratha-Korn Vilaichone.

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
