## [Decision Letter · Decision Letter 0]

30 Dec 2020

PONE-D-20-33347

Helicobacter pylori eradication improves motor fluctuations in advanced Parkinson's disease patients: A prospective cohort study (HP-PD trial)

PLOS ONE

Dear Dr. Lolekha,

Thank you for submitting your manuscript to PLOS ONE. After careful consideration, we feel that it has merit but does not fully meet PLOS ONE’s publication criteria as it currently stands. Therefore, we invite you to submit a revised version of the manuscript that addresses the points raised during the review process.

We look forward to receiving your revised manuscript.

Kind regards,

Wan-Long Chuang, M.D., Ph.D.

Academic Editor

PLOS ONE

Journal Requirements:

"We thank the Digestive Diseases Research Centre (DRC) at Thammasat University

Hospital and also Thai Otsuka Pharmaceutical Co., Ltd, who provided the infrared

spectrophotometer for the study."

"This study was supported by a grant from

Thammasat University Research Fund (TU Research Scholar, Contract No. 2/4/2562) and Thai Parkinson’s Disease and Movement Disorder Society (Thai-PDMDS). "

Additionally, because some of your funding information pertains to [commercial funding//patents], we ask you to provide an updated Competing Interests statement, declaring all sources of commercial funding.

In your Competing Interests statement, please confirm that your commercial funding does not alter your adherence to PLOS ONE Editorial policies and criteria by including the following statement: "This does not alter our adherence to PLOS ONE policies on sharing data and materials.” as detailed online in our guide for authors  http://journals.plos.org/plosone/s/competing-interests.  If this statement is not true and your adherence to PLOS policies on sharing data and materials is altered, please explain how.

Please include the updated Competing Interests Statement and Funding Statement in your cover letter. We will change the online submission form on your behalf.

Reviewers' comments:

Reviewer's Responses to Questions

**Comments to the Author**

1. Is the manuscript technically sound, and do the data support the conclusions?

Reviewer #1: Yes

Reviewer #2: Yes

2. Has the statistical analysis been performed appropriately and rigorously? 

Reviewer #1: Yes

Reviewer #2: Yes

3. Have the authors made all data underlying the findings in their manuscript fully available?

Reviewer #1: No

Reviewer #2: Yes

4. Is the manuscript presented in an intelligible fashion and written in standard English?

Reviewer #1: Yes

Reviewer #2: Yes

5. Review Comments to the Author

Reviewer #1: This is an interesting and important study. My comments are as follows:

1. The authors reported the changes of UPDRS motor scores and GI symptoms scores after H. pylori eradication (before and after). It is suggested that the authors may add a table to compare patients receiving eradication therapy versus those not receiving eradication therapy (HP negative group).

2. The case number of patients receiving eradication therapy is small (N=17), whereas the authors conducted multiple comparisons in the statistical analysis. Multiple testing correction might be needed or at least should be addressed in the discussion.

3. Whether the improvement of UPDRS score is directly related to the disappearance of H. pylori or alterations in gut microbiota may be addressed in the discussion section.

Reviewer #2: The article describes clearly about the effect of Helicobacter pylori (HP) eradication on the motor and non-motor performance in Parkinson's disease (PD). The authors concluded that HP eradication may increase "on" time and decrease "off" time and also significantly improved the end-of-dose score and GI symptoms including bloating and dysphagia. The article is well written with useful information. However, there are several questions needed to be clarified.

1.

Is there any difference of all the profiles in patients who failed the eradication therapy? Because PD patients had high placebo effects when receiving the treatment, the data may help clarify the effects of eradication therapy.

2.

Please provide the difference of LED and the percentage of relevant medications of PD patients with successful HP eradication with and without motor fluctuations. Because motor fluctuations may be improved by the adjustment of antiparkinsonian medications, the data may elucidate the effects of the eradication therapy.

3.

The International Movement Disorder Society has verified wearing-off questionnaires as WOQ-19 and WOQ-9, which are frequently used. The end-of-dose questionnaire in the article seemed to be similar to the questions of WOQ-9 or WO1-19. Could the authors explain the reasons why using the end-of-dose questionnaire? Please provide the all the profiles of questionnaires and the reference of the questionnaire.

4.

It’s very interesting and plausible to know the improvement of bloating symptoms in PD patients receiving eradication therapy. The authors mentioned there was no significant difference of motor scores and NMSQ scores, could the authors provide the evidence and explain the possible mechanism of the improvement of dysphagia symptoms?

6. PLOS authors have the option to publish the peer review history of their article (what does this mean?). If published, this will include your full peer review and any attached files.

Reviewer #1: No

Reviewer #2: No

---

## [Author Response · Author response to Decision Letter 0]

20 Jan 2021

I appreciated the constructive feedback of the editor and reviewers. I have addressed each of their concerns as outlined below.

Journal Requirements:

Response: Thank you for your suggestion. We have corrected the manuscript to meets PLOS ONE’s style requirements, including file naming and affiliations formatting.

For additional information about PLOS ONE ethical requirements for human subjects’ research, please refer to http://journals.plos.org/plosone/s/submission-guidelines#loc-human-subjects-research.

Response: We appreciate your comment. We have added this information in the methods section of the manuscript and ethics statement in the online submission information as the following paragraph. 

“All participants were verbally explained regarding the study before signing the consent form. Written informed consent was obtained from all subjects prior to enrolment.” 

"We thank the Digestive Diseases Research Centre (DRC) at Thammasat University

Hospital and also Thai Otsuka Pharmaceutical Co., Ltd, who provided the infrared

spectrophotometer for the study."

"This study was supported by a grant from

Thammasat University Research Fund (TU Research Scholar, Contract No. 2/4/2562) and Thai Parkinson’s Disease and Movement Disorder Society (Thai-PDMDS). "

Additionally, because some of your funding information pertains to [commercial funding//patents], we ask you to provide an updated Competing Interests statement, declaring all sources of commercial funding.

In your Competing Interests statement, please confirm that your commercial funding does not alter your adherence to PLOS ONE Editorial policies and criteria by including the following statement: "This does not alter our adherence to PLOS ONE policies on sharing data and materials.” as detailed online in our guide for authors http://journals.plos.org/plosone/s/competing-interests. If this statement is not true and your adherence to PLOS policies on sharing data and materials is altered, please explain how.

Please include the updated Competing Interests Statement and Funding Statement in your cover letter. We will change the online submission form on your behalf.

Response: Thank you for your suggestion. We have updated and removed funding-related text in the acknowledgment, funding statement, and competing interests as follows:

Acknowledgements

We thank the Digestive Diseases Research Centre (DRC) at Thammasat University Hospital for their assistance with the UBT tests. We would also like to show our gratitude to the patients and their families who took part and participated in the study.

Funding Statement

This study was supported by a grant from Thammasat University Research Fund (TU Research Scholar, Contract No. 2/4/2562) and Thai Parkinson’s Disease and Movement Disorder Society (Thai-PDMDS). The funders had no role in study design, data collection and analysis, decision to publish, or preparation of the manuscript.

Competing interests

The 13C-urea breath test and infrared spectrophotometer used in the study was provided by Thai Otsuka Pharmaceutical Co., Ltd. This does not alter our adherence to PLOS ONE policies on sharing data and materials. 

Response: Thank you for your suggestion. We have updated captions for supporting information files at the end of the manuscript. 

Reviewers' comments:

Reviewer's Responses to Questions

Comments to the Author

1. Is the manuscript technically sound, and do the data support the conclusions?

Reviewer #1: Yes

Reviewer #2: Yes

2. Has the statistical analysis been performed appropriately and rigorously? 

Reviewer #1: Yes

Reviewer #2: Yes

3. Have the authors made all data underlying the findings in their manuscript fully available?

The PLOS Data policy requires authors to make all data underlying the findings described in their manuscript fully available without restriction, with rare exception (please refer to the Data Availability Statement in the manuscript PDF file). The data should be provided as part of the manuscript or its supporting information or deposited to a public repository. For example, in addition to summary statistics, the data points behind means, medians and variance measures should be available. If there are restrictions on publicly sharing data—e.g. participant privacy or use of data from a third party—those must be specified.

Reviewer #1: No

Reviewer #2: Yes

Response: Thank you for your suggestion. We have updated tables and data underlying the findings in the manuscript as suggestion. 

4. Is the manuscript presented in an intelligible fashion and written in standard English?

Reviewer #1: Yes

Reviewer #2: Yes

5. Review Comments to the Author

Reviewer #1: This is an interesting and important study. My comments are as follows:

1. The authors reported the changes of UPDRS motor scores and GI symptoms scores after H. pylori eradication (before and after). It is suggested that the authors may add a table to compare patients receiving eradication therapy versus those not receiving eradication therapy (HP negative group).

Response: We appreciate your important suggestion. We have compared general demographic data of the study population between HP positive (all have received eradication therapy) and HP negative group in Table 1. Also, we compared the successful HP eradication group and HP negative group's clinical symptoms in Table 4 as your suggestion.

2. The case number of patients receiving eradication therapy is small (N=17), whereas the authors conducted multiple comparisons in the statistical analysis. Multiple testing correction might be needed or at least should be addressed in the discussion.

Response: We appreciate your important comment, and we are concerned about this issue. We have performed multiple testing correction by Bonferroni adjustment in our statistic data and revealed the statistically significant difference in total TWOQ-9 score before and after eradication therapy. We have mentioned this issue in the results section and also addressed this limitation in the discussion as follows:

Results

“Multiple testing correction by Bonferroni adjustment showed a statistically significant improvement in total TWOQ-9 score.”

Discussion section 

“We acknowledge that our present study has some limitations. First, our study was based on one center and the number of participants was limited.……. Also, there were multiple comparisons in the statistical analysis that might cause statistical significance occurred by chance.” 

3. Whether the improvement of UPDRS score is directly related to the disappearance of H. pylori or alterations in gut microbiota may be addressed in the discussion section.

Response: We appreciate your important comment. There is emerging evidence to support the hypothesis that gut microbiota alteration may be related to the pathogenesis, clinical symptoms, and PD progression. The compositional changes of microbiota may occur after antibiotic usage, smoking, or dietary modifications. In our study, the alterations in microbiota may be occurred after receiving the HP eradiation therapy and might relate to the clinical improvement or L-dopa absorption. Since our study did not aim or provide the baseline type and number of gut microbiota, we could not answer this question. We have addressed this issue in our limitation in the discussion as follows:

“Lastly, benefits seen in our short-term study may not translate into improvements in the longer term, due to variations in lifestyle factors such as dietary habits, the compositional changes of gut microbiota, and other associated factors that might effect on the disease progression or interfere with L-dopa absorption.”

Reviewer #2: The article describes clearly about the effect of Helicobacter pylori (HP) eradication on the motor and non-motor performance in Parkinson's disease (PD). The authors concluded that HP eradication may increase "on" time and decrease "off" time and also significantly improved the end-of-dose score and GI symptoms including bloating and dysphagia. The article is well written with useful information. However, there are several questions needed to be clarified.

1. Is there any difference of all the profiles in patients who failed the eradication therapy? Because PD patients had high placebo effects when receiving the treatment, the data may help clarify the effects of eradication therapy.

Response: We appreciate your comment. We have added the clinical profiles of patients who fail to eradicate therapy in table 3 as your suggestion.

Besides, we also added the demographic profiles of patients who fail the eradication therapy in Table 5. Patients who failed the eradication therapy were all male and had a significantly lower intake of vegetables and fruit (1.4 vs. 2.4 servings/day, p < 0.01) as well as a higher rate of smoking (40.0 vs. 0.0%, p < 0.01) compared with the HP eradication group. There was no statistically significant difference in age, age at onset, disease duration, SE-ADL scale, H&Y stage, UPDRS motor score, LED and antiparkinsonian medications between groups (Table 5).

2. Please provide the difference of LED and the percentage of relevant medications of PD patients with successful HP eradication with and without motor fluctuations. Because motor fluctuations may be improved by the adjustment of antiparkinsonian medications, the data may elucidate the effects of the eradication therapy.

Response: Thank you for your comment. All patients in the study had motor fluctuations. All patients who received the HP eradication therapy continued their anti-PD medications and were not allowed to adjust or change their PD medication regimen (mentioned in methods: the HP eradication therapy and outcome assessment section). Therefore, there is no change in LED and the percentage of PD medication before and after receiving HP eradication therapy. We believed that the outcomes in our study were mainly the effects of eradication therapy. However, there is no control of the variations in lifestyle factors such as dietary habits or non-parkinsonian medications that might interfere with L-dopa absorption (mentioned in the discussion).

3. The International Movement Disorder Society has verified wearing-off questionnaires as WOQ-19 and WOQ-9, which are frequently used. The end-of-dose questionnaire in the article seemed to be similar to the questions of WOQ-9 or WO1-19. Could the authors explain the reasons why using the end-of-dose questionnaire? Please provide the all the profiles of questionnaires and the reference of the questionnaire.

Response: We appreciate your comment and concern. The end-of-dose (EOD) questionnaire that we used in our article/center is similar to the questions of WOQ-9 in Thai language. The questionnaire was developed by Thai Parkinson’s Disease and Movement Disorder Society (Thai PDMDS) and was published in Thai clinical practice guideline for diagnosis and management of PD in 2012. For clarification and understanding, we had edited the name of the questionnaire in our article from “the end-of-dose questionnaire” to "Thai version of the 9-item Wearing-off Questionnaire (TWOQ-9)". We also provided a questionnaire in the supporting file (S1 Study Protocol Thai), data finding in Table 1 and 2, and the article's reference.

4. It’s very interesting and plausible to know the improvement of bloating symptoms in PD patients receiving eradication therapy. The authors mentioned there was no significant difference of motor scores and NMSQ scores, could the authors provide the evidence and explain the possible mechanism of the improvement of dysphagia symptoms?

Response: Thank you very much for your comment. We have added the possible mechanism in the discussion section as follows:

“Dysphagia typically emerges in PD patients with advanced motor symptoms, and often related to gastroesophageal reflux disease. Eradication of HP in our study did not significantly improve motor scores or reflux symptoms. However, HP positive patients who received eradication treatment demonstrated significant improvement of dysphagia subscore. Also, the post-eradication total GI symptoms, bloating, heartburn, and dysphagia subscores were significantly less than baseline scores of HP negative patients. This might suggest additional effects of eradication therapy, especially from a gastric acid-lowering agent on gastroesophageal reflux disease that might contribute to dysphagia in PD patients. It should be noted that patients with severe dysphagia were excluded from the study. The significant improvement of dysphagia score in our study was minimal and might not significantly be observed in clinical practice.” 

6. PLOS authors have the option to publish the peer review history of their article (what does this mean?). If published, this will include your full peer review and any attached files.

Do you want your identity to be public for this peer review? For information about this choice, including consent withdrawal, please see our Privacy Policy.

Reviewer #1: No

Reviewer #2: No

Response: Thank you very much for your suggestion. We have edited figure files using PACE digital diagnostic tool as your suggestion. 

Best regards,

---

## [Decision Letter · Decision Letter 1]

9 Mar 2021

PONE-D-20-33347R1

Helicobacter pylori eradication improves motor fluctuations in advanced Parkinson's disease patients: A prospective cohort study (HP-PD trial)

PLOS ONE

Dear Dr. Lolekha,

Thank you for submitting your manuscript to PLOS ONE. After careful consideration, we feel that it has merit but does not fully meet PLOS ONE’s publication criteria as it currently stands. Therefore, we invite you to submit a revised version of the manuscript that addresses the points raised during the review process.

We look forward to receiving your revised manuscript.

Kind regards,

Wan-Long Chuang, M.D., Ph.D.

Academic Editor

PLOS ONE

Journal Requirements:

Reviewers' comments:

Reviewer's Responses to Questions

**Comments to the Author**

1. If the authors have adequately addressed your comments raised in a previous round of review and you feel that this manuscript is now acceptable for publication, you may indicate that here to bypass the “Comments to the Author” section, enter your conflict of interest statement in the “Confidential to Editor” section, and submit your "Accept" recommendation.

Reviewer #1: All comments have been addressed

Reviewer #2: All comments have been addressed

Reviewer #3: (No Response)

2. Is the manuscript technically sound, and do the data support the conclusions?

Reviewer #1: Yes

Reviewer #2: Yes

Reviewer #3: Yes

3. Has the statistical analysis been performed appropriately and rigorously? 

Reviewer #1: Yes

Reviewer #2: Yes

Reviewer #3: Yes

4. Have the authors made all data underlying the findings in their manuscript fully available?

Reviewer #1: Yes

Reviewer #2: Yes

Reviewer #3: No

5. Is the manuscript presented in an intelligible fashion and written in standard English?

Reviewer #1: Yes

Reviewer #2: Yes

Reviewer #3: Yes

6. Review Comments to the Author

Reviewer #1: This is an interesting and important study. The authors have responded to the comments from the reviewer.

Reviewer #2: The great efforts of the authors to answer the question are appreciated. I think their work may help the clinicians and scientists to uncover the complex mechanism between HP and PD.

Reviewer #3: Abstract: There are two percentages mentioned, and it isn't quite clear how these are calculated - 22 patients (55%) and 17 patients (77.3%) - the higher number has the lower percentage, so, to be correct the denominator must be different, but it isn't clear what it is from this text. Could you please clarify, perhaps by stating what the denominator was (22/40 and then 17/22?)?

Study design - pg 3, line 74 - what was the effect size or mean and standard deviation of change in daily on-off time, L-dopa onset or peak time used in the power calculation? What was the difference in groups the study aimed to test? or was it only in those patients who eradicated HP that were of interest? Why were the patients who were HP negative at baseline included? How were the numbers reduced from 163 to 40?

It is unclear how the various groups were recruited from the text, but it is clearer once Figure 1 is examined - could the text be cleared up?

pg 9, line 190 - mention multiple testing correction method with statistical methods (as "p<0.05 was considered to be statistically significant") on page 6 isn't the full story. Include the Bonferroni corrected p-values where they were calculated.

Table 4: It needs to be clear that this is baseline HP negative results and after treatment HP eradication results. For the statistically significant results - it may have been that the results were just different at baseline between these two groups of patients - would it have been of interest to see if there was a difference in the later time point data for the two groups, adjusted for the baseline variables?

Could be 4 columns of results be presented? Pre and post treatment for those who were HP negative at baseline and those who ended up as HP negative after treatment?

7. PLOS authors have the option to publish the peer review history of their article (what does this mean?). If published, this will include your full peer review and any attached files.

Reviewer #1: No

Reviewer #2: No

Reviewer #3: No

---

## [Author Response · Author response to Decision Letter 1]

25 Mar 2021

Reviewer #3: 

1. Abstract: There are two percentages mentioned, and it isn't quite clear how these are calculated - 22 patients (55%) and 17 patients (77.3%) - the higher number has the lower percentage, so, to be correct the denominator must be different, but it isn't clear what it is from this text. Could you please clarify, perhaps by stating what the denominator was (22/40 and then 17/22?)?

Response: We appreciate your comment. For clarification, we have edited the abstract as follow:

“A total of 163 PD patients were assessed, of whom 40 were enrolled. Fifty-five percent of the enrolled patients (22/40) had a current HP infection, whereas HP eradication was identified in 17 of 22 (77.3 %) patients who received eradication therapy.”

2. Study design - pg 3, line 74 - what was the effect size or mean and standard deviation of change in daily on-off time, L-dopa onset or peak time used in the power calculation? What was the difference in groups the study aimed to test? or was it only in those patients who eradicated HP that were of interest? Why were the patients who were HP negative at baseline included? 

How were the numbers reduced from 163 to 40? It is unclear how the various groups were recruited from the text, but it is clearer once Figure 1 is examined - could the text be cleared up?

Response: We appreciate your important comments, and we have addressed each comment as follows:

- What was the effect size or mean and standard deviation of change in daily on-off time, L-dopa onset or peak time used in the power calculation?

The effect size we used in our study was set at 0.8. From previous studies, the mean change in L-dopa onset time and total daily on time after eradication therapy were approximately 13 and 106 minutes, respectively. There was no report of the standard deviation of the mean differences. We have revised the paragraph as below:

“The estimated sample size was determined according to the previous studies [11–14] with a level of significance of 5%, the statistical power of 80%, and the effect size of 0.8. The calculated number of HP infected patients needed in this study was 15 patients. Given the prevalence of HP infection in general Thai population of 45% and the possibility of a 15% drop out rate, a total of 40 advanced PD patients were recruited.”

- What was the difference in groups the study aimed to test? or was it only in those patients who eradicated HP that were of interest?

The study aimed to compare the clinical effects between before and after receiving HP eradication therapy in advanced PD, so it was mainly focused on PD patients with current HP infection. 

- Why were the patients who were HP negative at baseline included?

 To identify current HP infected patients, we needed to screen the HP status in all advanced PD. We included the HP negative patients in order to identify risk factors or clinical effects that might be associated with HP infection in PD patients. As shown in Table 1, the young age and history of anticholinergic use were risk factors for HP infection. Also, the HP infected patients demonstrated significant delayed L-dopa onset time and tremor during the off period than the non-infected group. These results might support the possible benefit of HP eradication therapy in the later part of our study. 

Moreover, there was no significant difference in motor symptoms (UPDRS motor score, daily on-off time, quality of life score) between HP negative patients at baseline and the successful HP eradication therapy group (table 4). These results might suggest that HP eradication is only beneficial in the treatment of motor fluctuations by increased daily on time but not motor symptoms or affects disease progression.

- How were the numbers reduced from 163 to 40? It is unclear how the various groups were recruited from the text, but it is clearer once Figure 1 is examined - could the text be cleared up?

 Thank you for your suggestion, we have revised the recruitment process as follow:

“A total of 163 PD patients were consecutively assessed for eligibility. Of these, 123 patients were excluded, and 40 patients were enrolled in the study (Fig 1).” 

3. pg 9, line 190 - mention multiple testing correction method with statistical methods (as "p<0.05 was considered to be statistically significant") on page 6 isn't the full story. Include the Bonferroni corrected p-values where they were calculated.

Response: Thank you very much for your comment. We did the corrected p-values according to the reviewer’s suggestion in the revised manuscript, and we found that only the total TWOQ-9 score was statistically significant with the corrected p = 0.029. We have added the corrected p-values as the following paragraph and provide data of corrected p-values in the supporting table (S1 Table).

“Multiple testing correction by Bonferroni adjustment showed a statistically significant improvement only in total TWOQ- 9 score (corrected p = 0.029) (S1 Table).”

4. Table 4: It needs to be clear that this is baseline HP negative results and after treatment HP eradication results. For the statistically significant results - it may have been that the results were just different at baseline between these two groups of patients - would it have been of interest to see if there was a difference in the later time point data for the two groups, adjusted for the baseline variables? Could be 4 columns of results be presented? Pre and post treatment for those who were HP negative at baseline and those who ended up as HP negative after treatment?

Response: Thank you very much for your suggestion. We have revised Table 4 as below. Patients with HP negative were not reassessed at the 6-week follow-up, so we compared pre-post HP eradication with the HP negative at baseline. From table 4, a significant improvement of the GI symptom scores was observed only in the post-HP eradication. Also, the TWOQ-9 score and L-dopa onset time after the eradication were comparable to the HP negative at baseline.

---

## [Decision Letter · Decision Letter 2]

19 Apr 2021

Helicobacter pylori eradication improves motor fluctuations in advanced Parkinson's disease patients: A prospective cohort study (HP-PD trial)

PONE-D-20-33347R2

Dear Dr. Lolekha,

We’re pleased to inform you that your manuscript has been judged scientifically suitable for publication and will be formally accepted for publication once it meets all outstanding technical requirements.

Kind regards,

Wan-Long Chuang, M.D., Ph.D.

Academic Editor

PLOS ONE

Additional Editor Comments (optional):

Reviewers' comments:

Reviewer's Responses to Questions

**Comments to the Author**

1. If the authors have adequately addressed your comments raised in a previous round of review and you feel that this manuscript is now acceptable for publication, you may indicate that here to bypass the “Comments to the Author” section, enter your conflict of interest statement in the “Confidential to Editor” section, and submit your "Accept" recommendation.

Reviewer #3: All comments have been addressed

2. Is the manuscript technically sound, and do the data support the conclusions?

Reviewer #3: (No Response)

3. Has the statistical analysis been performed appropriately and rigorously? 

Reviewer #3: (No Response)

4. Have the authors made all data underlying the findings in their manuscript fully available?

Reviewer #3: (No Response)

5. Is the manuscript presented in an intelligible fashion and written in standard English?

Reviewer #3: (No Response)

6. Review Comments to the Author

Reviewer #3: (No Response)

7. PLOS authors have the option to publish the peer review history of their article (what does this mean?). If published, this will include your full peer review and any attached files.

Reviewer #3: No

---

## [Editor Report · Acceptance letter]

26 Apr 2021

PONE-D-20-33347R2 

*Helicobacter pylori* eradication improves motor fluctuations in advanced Parkinson's disease patients: A prospective cohort study (HP-PD trial) 

Dear Dr. Lolekha:

I'm pleased to inform you that your manuscript has been deemed suitable for publication in PLOS ONE. Congratulations! Your manuscript is now with our production department. 

Kind regards, 

on behalf of

Dr. Wan-Long Chuang 

Academic Editor

PLOS ONE